# Quality of aerobic training description and its relation to intervention efficacy in chronic obstructive pulmonary disease trials: study protocol for a systematic review, meta-analysis and meta-regression

Johan Jakobsson ,[1] Anouk A F Stoffels,[2] Hieronymus W H van Hees,[2] Jana De Brandt,[1] André Nyberg,[1] Peter Klijn[3,4]

JJ and AAFS are joint first authors.

AN and PK are joint senior authors.

For numbered affiliations see end of article.

**Correspondence to**
Johan Jakobsson;
johan.jakobsson@umu.se

## ABSTRACT

**Introduction** Chronic obstructive pulmonary disease (COPD) is a major global health concern, characterised by ventilatory constraints, decreased cardiovascular fitness and reduced limb muscle function, profoundly affecting patients' quality of life. Aerobic training plays a crucial role in the treatment of COPD, but the variability in methodologies and incomplete reporting of key components in aerobic training trials limits the assessment of their effectiveness. This systematic review aims to critically evaluate the application of training principles and reporting of key components in aerobic training trials in randomised controlled trials (RCTs) in the COPD literature.

**Methods and analysis** The protocol adheres to the Preferred Reporting Items for Systematic reviews and Meta-Analyses Protocol guidelines. The review will include RCTs utilising aerobic training in individuals with COPD. A comprehensive search, following a predefined search strategy will identify studies published from 2007 to 2024 in English from MEDLINE, Embase, CINAHL, CENTRAL and PEDro. Studies including people with COPD and any aerobic training intervention will be included. Two reviewers will independently screen abstracts and titles for inclusion. Two reviewers will independently conduct the screening of full-text documents and data extraction. Study quality will be assessed using the Tool for the assESsment of sTudy quality and bias in Exercise, specifically developed for exercise training studies. The certainty of the evidence will be evaluated using the Grading of Recommendations Assessment, Development and Evaluation approach. A systematic synthesis will be provided, with meta-analyses and meta-regression when appropriate.

**Ethics and dissemination** As this review will involve the analysis of published data, ethical approval is not required. The findings of this review will be disseminated through peer-reviewed publications and conference presentations.

**PROSPERO registration number** CRD42021247343.

## STRENGTHS AND LIMITATIONS OF THIS STUDY

⇒ The protocol includes a comprehensive search strategy and broad inclusion criteria to allow for a comprehensive synthesis of available evidence.
⇒ This protocol is reported according to the Preferred Reporting Items for Systematic reviews and Meta-Analyses Protocols.
⇒ We will examine the quality of included studies using the Tool for the assESsment of sTudy quality and bias in EXercise specifically developed for trials with exercise interventions.
⇒ Exclusion of literature written in languages not known by the research group might leave relevant literature out of the systematic review.

## INTRODUCTION

Chronic obstructive pulmonary disease (COPD) stands as the third leading cause of death,[1] with more than 384 million individuals diagnosed globally.[2] In the management of COPD, the beneficial impacts of pulmonary rehabilitation (PR), incorporating aerobic exercise training as a key element, are well documented.[2–5] These improvements typically manifest in increased cardiorespiratory fitness,[6] better overall quality of life and it reduces the burden of symptoms.[3 7–9] In fact, among different exercise modalities within PR, aerobic training is the most frequently applied and investigated mode of exercise in COPD,[3 10] often investigated using a randomised controlled trial (RCT).

However, the reporting of RCTs is often incomplete, giving rise to the Consolidated Standards of Reporting Trials (CONSORT) statement in 1996.[11] Provided that the field of medicine depends on transparent reporting

BMJ

of RCTs,[12] inadequate reporting of trials is troublesome. As a result, the reliability and validity of findings cannot be judged, nor data synthesised for systematic reviews and meta-analyses.[13] Of course, the literature of rehabilitation research and exercise training is not an exception from this.[14] Indeed, exercise training interventions are often poorly described,[15] and the field suffers from other concerns regarding methodological rigour and problems with replication.[16] This negatively influences the quality of evidence, as recently concluded by a review investigating methodological rigour and reporting practices in rehabilitation research.[14] In addition, it also contributes to heterogeneity between PR programmes and hinders international benchmarking.

Inadequate reporting practices of exercise interventions led to the development of the Consensus on Exercise Reporting Template (CERT) in 2016.[15] As previously stated, it is almost unethical that researchers are still able, almost without any regulation, to design and test exercise interventions that likely have a low potential for being effective.[17] Adequate reporting of research, including exercise interventions, is the first step to help the reader to understand what the researcher did, assess intervention quality and effectiveness, translate research findings into clinical practice and synthesise data in systematic reviews and meta-analyses for evidence-based guidelines. Although the efficacy of aerobic training interventions in COPD is evident, we, and others,[18 19] have acknowledged a high level of heterogeneity in exercise training protocols and methodological rigour, as well as lacking descriptions of exercise interventions in the COPD literature. This ambiguity complicates the assessment of intervention effectiveness and challenges the replicability of these interventions and their applicability into clinical settings. As such, complete reporting of training interventions is essential. Inadequate reporting of aerobic training interventions has been reviewed previously in other chronic diseases,[20–24] while trials in the COPD literature remains to be investigated.

Specifically, for exercise training trials, it is important to report all factors influencing the response to training. First, the well-established training principles[25–27] (table 1) should be applied in designing an exercise programme to ensure the most appropriate exercise type and dose to achieve the desired outcome. Second, important exercise training variables[15] that contribute to the training stimulus should be reported. These components should cover the FITT-principles (frequency, intensity, time, type) and information on rest, sequence of exercises and programme duration.[15] Evidence on the importance of incorporating the FITT-principles within aerobic training studies are starting to emerge. Recent data suggest that FITT-principles such as frequency and time seem to be associated with an increase in peak oxygen uptake following aerobic training in individuals with COPD.[6] Third, items concerning the external validity, such as eligibility criteria, setting, thorough reporting of participant characteristics, supervision and adherence

to interventions must also be described.[12 15 17 28] Still, whether the efficacy of aerobic training interventions among people with COPD is related to the application of exercise training principles and/or a more complete description of key exercise training variables remains to be fully determined.

### Aims and objectives

We aim to investigate the application of exercise training principles and completeness of reporting in aerobic training interventions in the COPD literature. We also aim to investigate if efficacy of the interventions provided is related to the completeness of reporting and application of training principles. Our specific research questions are as follows:

'To what extent does RCTs with an aerobic training intervention in the COPD literature apply and report the principles of exercise training, and how well are the key training variables, intervention and participant characteristics reported in these trials?'

'Is the efficacy of aerobic training interventions related to the application of exercise training principles and/ or a more complete description of key exercise training variables?'

## METHODS AND ANALYSIS

The study protocol of this systematic review is reported in accordance with the Preferred Reporting Items for Systematic reviews and Meta-Analyses Protocols (PRISMA-P) guideline (provided in online supplemental appendix 1). The final review will be reported according to the PRISMA 2020 guidelines.[29]

### Eligibility criteria

We will select studies according to the criteria below.

### Study designs

Only RCTs will be included in this systematic review. Other longitudinal study designs, such as controlled non-randomised studies of interventions, cluster trials and prospective comparative cohort studies will be excluded. Studies with a cross-sectional design, retrospective studies, case series, case reports, commentaries and systematic or narrative reviews, will be excluded.

### Participants

Any eligible study design with people with COPD being ≥90% of the participants will be considered. Studies with a higher proportion of non-COPD people can be included if the data for the COPD group is, or can be, separated. The COPD diagnosis should be based on a forced expiratory volume in one second ($FEV_1$) to forced vital capacity (FVC)<0.7. There will be no eligibility criteria for disease severity, sex, age, body composition or ethnicity. Participants with additional comorbidities will not be excluded, but COPD should be the primary disease of the participants.

## Interventions

Any type of aerobic training intervention of the lower limbs which is aerobically demanding is of interest. For example, cycling, walking, swimming, water-based movements and aerobics. There will be no eligibility criteria for duration of the intervention or number of exercise sessions.

## Comparators

Since this study does not intend to compare the effects between two different interventions, there will be no eligibility criteria for comparators. If there is one group with an eligible aerobic training intervention, it will be included independent of type of, or the existence of, a control group in this study.

## Outcomes

During the screening phase, studies will not be excluded based on outcomes. One objective of this systematic review is to investigate whether the key training principles are applied, and whether training components are described completely. See 'data collection' for which data that will be extracted for this objective.

For the second objective, the following outcomes are eligible.

### Primary outcomes

► 6 min walking distance (6MWD metres).
► Incremental shuttle walk test (ISWT, metres)
► Endurance shuttle walk test (ESWT, duration)
► Peak work rate (watts) or $VO_2max$ (mL/min/kg) during cardiopulmonary exercise test (CPET).
► Constant work-rate test (CWRT, duration)

### Secondary outcomes

► Short physical performance battery (SPPB, score)
► Sit-to-stand tests: 5-time sit-to-stand, 30 s sit-to-stand or 1 min sit-to-stand (time or reps)
► Timed-up and go test (TUG; s).
► Stair climb power test (watt).
► Steep ramp test (watt).

## Timing

Data from before ('baseline' or 'pre'), immediately after the intervention ('post') and if applicable, additional follow-ups, will be included for intervention studies with more than two time points of outcome assessments.

## Setting

There will be no eligibility criteria for type of setting. Examples of different settings could be home-based, hospital-based, inpatient, outpatient, community based or a combination of these.

## Timespan

Articles from 1 January 2007 until 2024 will be included. An initial search was done in May 2021, which will be updated prior to completion of the review. This timespan was chosen since it would include RCTs published

after the first joint statement on PR[30] by the American Thoracic Society and the European Respiratory Society in 2006.

## Language

Only articles written in English will be included. A list of possibly relevant titles in other languages will be provided.

## Search methods for identification of studies
### Electronic searches

– Studies will be identified from:
– MEDLINE (PubMed interface)
– Embase
– CINAHL
– Cochrane Central Register of Controlled Trials (CENTRAL)
– PEDro

### Other resources

The reference list of includes studies will be hand-searched for other relevant studies. Also, other relevant systematic and narrative review studies will be checked for additional studies eligible for inclusion. Conference proceedings will not be included. No grey literature will be searched since we believe it will not add any additional body of literature.

The Cochrane Database of Systematic Reviews and PROSPERO have been searched for existing or ongoing reviews on the topic, with no findings.

## Search strategy

To identify relevant studies, a comprehensive search will be performed. The search strategy has been developed by the authors who are researchers in respiratory physiotherapy and clinical exercise physiology, together with a librarian of the Radboud University (Nijmegen, The Netherlands). The original search strategy was developed for PubMed and has been adapted to other databases' syntax and subject headings. Comprehensive searches were constructed of both index terming (medical subject headings term), free-text terms and synonyms. An English language restriction was imposed on the search. The search strategy for the electronic searches, initially done in May 2021, but which will be updated in 2024, is presented in the online supplemental appendix 2.

## Data collection and analysis
### Study records

The results from the literature searches will be uploaded to Rayyan,[31] a web-tool for collaborative systematic reviews. Screening question for selection will be pilot tested on a few papers.

The search process will be documented, including:

► the name of the database searched.
► the name of the database provider/system used.
► the date when the search was run.
► the years covered by the search.
► the search terms used, hits per search term and number of articles retrieved.

## Selection process

Titles and abstracts retrieved from the searches and additional sources will be reviewed independently by two reviewers to identify potentially relevant studies for inclusion based on the eligibility criteria. Studies will be coded as 'yes', 'no' or 'maybe'. Studies with 'no' from both reviewers will be directly excluded. Studies with two 'yes' will be directly included to the next stage. Studies with conflicting decisions will be checked by at least one reviewer for inclusion or exclusion. Then, we will retrieve the full texts of all potentially eligible studies. If needed, authors will be contacted to resolve questions regarding eligibility.

Full-text articles of included studies will be independently screened by pairs of reviewers for eligibility. A third reviewer will assess agreement and resolve disagreements on eligibility using a majority rule. Neither of the review authors will be blind to the journal titles, the study authors, or institutions.

The results of the selection process will be documented reported in sufficient detail to complete a PRISMA flow diagram. Missing papers will be requested from study authors using e-mail. Interassessor agreement will be assessed using Kappa statistics.

## Data collection process

From each included study, two reviewers will independently extract information on study characteristics, participants characteristics, intervention descriptions and outcome data. Data will be entered into a pilot-tested data extraction form developed a priori. All the extracted variables will then be used in the synthesis and quality appraisal of the aerobic training description. A third reviewer will assess agreement between data extractors and resolve disagreements using a majority rule. If needed, a fourth reviewer will be consulted.

The data as outlined below will be extracted. First, it will be reported whether the data are reported ('1') or not reported ('0'), if its insufficiently reported, not clear or not applicable (NA). Second, the actual data will be extracted, such as the corresponding data value or a text description. Continuous data will be reported as mean±SD or median (Q1–Q3). If it is reported otherwise, such as mean±SEM, this will be noted. The following variables will be extracted:

► Study details: first author name, publication year, title, journal, study design, study objective, inclusion criteria, exclusion criteria and country of execution. If country of execution is not mentioned in the text, it will be derived from where ethical permission have been obtained.

Participants characteristics will be extracted and reported per intervention group. The aerobic training group will always be coded as the intervention group, and any other group will be coded as control. If the study consists of, and is reported in several subgroups, such as per disease severity, data will be reported per subgroup.

Training principles, reporting of participant characteristics, intervention characteristics and training variables will be extracted for our first objective. In addition, outcomes will be extracted for our second objective.

Participant characteristics that will be extracted:
► General
  ○ Sample size (n)
  ○ Age (years)
  ○ Sex distribution (number)
► Pulmonary function (coded as reported if any of the data below is provided):
  ○ $FEV_1$ (L).
  ○ $FEV_1$ (% predicted).
  ○ GOLD stages I, II, III or IV (number or % per group).
  ○ GOLD stage A, B, C or D (or A, B or E when applicable) (number or % per group).
  ○ $FEV_1/FVC$ (ratio).
  ○ RV/TLC (ratio).
  ○ Dynamic hyperinflation (change in inspiratory capacity, % or L).
  ○ DLco (mL/min/kPa).
  ○ DLco (% predicted).
► Body composition (coded as reported if any of the data below are provided):
  ○ Body mass index ($kg/m^2$).
  ○ Fat free mass index ($kg/m^2$)
  ○ Fat mass index ($kg/m^2$)
  ○ Muscle mass (kg)
► Muscle strength (coded as reported if any of the data below are provided):
  ○ Muscle strength (kg, Nm or as appropriate)
  ○ Muscle strength (% predicted, with reference to predicted values)
  ○ Text description of apparatus and type of contraction used in the measurement.
► Exercise capacity: (coded as reported if any of the data below are provided):
  ○ CPET ($VO_2$ peak mL/min/kg and % predicted, peak watt (W) and % predicted).
  ○ ISWT (metres, % predicted)
  ○ 6-MWD (metres, % predicted).
► Exercise tolerance: (coded as reported if any of the data below are provided):
  ○ CWRT (duration).
  ○ ESWT (duration)
► Training status: text description regarding previous experience or training status of participants
► Physical activity (CERT item 15): text description (ie, sedentary and active) or as steps per day, the physical activity level or metabolic equivalents.
► Smoking status: average packyears and number of participants per group of never smokers, smokers, former smokers, and non-smokers.
► Comorbidities: frequency per comorbidity, Charlson-comorbidity index or hospital anxiety and depression scale.
► Specific phenotype: text description of specific COPD-phenotypes, for example, frequent exacerbator,

alpha-1-antitrypsine deficiency, asthma-COPD-overlap, among others.

Aerobic training intervention characteristics that will be extracted:

► Aerobic training objective: rationale regarding the benefits of the exercise programme.
► Training device used, including manufacturer and type (CERT item 1). For example, only 'cycle ergometer' is insufficient reporting, as type and manufacturer is missing.
► Type of exercise (CERT item 13): for example, walking, cycling, rowing stair walking.
► Adjuncts (CERT item 10): for example, other exercise modalities, breathing exercises, nutrition interventions, medications or education.
► Description of control intervention (CERT item 9)
► Place of aerobic training in intervention: standalone, part of PR, as an adjunct or other.
► Supervision (CERT item 2, 4 16a): type of supervision and ratio between supervisors and participants. Only information whether it was supervised or not is insufficient.
► Supervision qualifications: experience, skills, and education of supervisors.
► Setting (CERT item 12): inpatient, outpatient, home-based, community-based, hospital based, combination or other.
► Delivery (CERT item 3): group or individual. If the delivery is in group, the group size should be specified.
► Attendance: number of, or % of, attended sessions.
► Adherence to protocol (CERT item 5, 13, 14, 16b): in addition to the number or percentage of attended sessions, information should be provided on the adherence to the prespecified protocol. Thus, whether performed intensity, duration and total load of the training was performed as planned.
► Safety: information regarding the number and type of adverse events.
► Dropouts: information regarding the number reason for drop-out.

Training principles that will be extracted are described in table 1.

We have chosen to not include the principle *diminishing returns* in our critical appraisal. We argue that his principle, which states *that the expected degree of improvement in fitness decreases as individuals become more fit, thereby increasing the effort required for further improvements*, is highly difficult to distinctly judge as 'yes' or 'no', separately from other principles, specifically progression and overload. For example, if both progression and overload is applied, we believe diminishing returns is considered.

Training variables that will be extracted (extended version of CERT item 13):

► Programme duration (weeks).
► Programme frequency (sessions/week).
► Exercise intensity (ie, watts or km/hour).
► Work-phase duration (min: s).

► Rest periods between series (min: s): only applicable for interval training with rest periods, or continuous training with rest periods are allowed.
► Type of rest between series, if applicable (active and passive).
► Intensity of active rest between series, if applicable (ie, watts or km/hour).
► Rest periods between exercises (min: s), applicable on rest periods between exercises which are part of the training, including other components than the aerobic training.
► Sequence of exercises, if applicable.

Outcomes that will be extracted (reported as yes or no and corresponding data):

Primary outcomes

► 6MWD (metres).
► ISWT (metres).
► ESWT (time).
► Peak work rate (watts) or $VO_2max$ (mL/min/kg) during CPET.
► Constant work-rate test (time, min: s).

Secondary outcomes

► SPPB, score.
► Sit-to-stand tests: 5-time sit-to-stand, 30 s sit-to-stand or 1 min sit-to-stand (time or reps)
► TUG (s).
► Stair climb power test (Watt)
► Steep ramp test (Watt)

We consider outcomes related to cardiorespiratory capacity and exercise tolerance primary given that these are vital objectives in international COPD treatment guidelines.[3 28 32]

## Risk of bias and study quality assessment

The risk of bias and reporting in eligible studies will be assessed independently by two reviewers. Disagreements will be resolved by consensus or including a third reviewer. No study will be excluded due to high risk of bias, but sensitivity analyses excluding poor-quality studies will be performed. The risk of bias assessment applies to our secondary objective describe in this protocol. Thus, the risk of bias assessment will be presented along with the results regarding the secondary objective.

To assess risk of bias, we will use the TESTEX (Tool for the assESsment of sTudy quality and bias in EXercise) tool. TESTEX[33] is specifically designed for evaluating the risk of bias and reporting in exercise training studies. The TESTEX contains 12 measures: five on study quality and seven on study reporting. A total of 15 points can be provided, where higher ratings mean higher quality (lower risk of bias). While no validated cut-off scores are provided for TESTEX, we will categorise the studies according to the average score as 'high quality' (≥12 points), 'good quality' (7–11 points) and 'low quality' (≤6 points) as previously described.[34]

Each study will be independently assessed by two reviewers, and discrepancies will be resolved through discussion or consultation with a third reviewer if needed.

**Table 1**  Exercise training principles

| Principle | Criteria used in this review | Examples and elaboration |
|---|---|---|
| *Progression*: over time, adaptations in the physiological component being trained occur. Progression refers to the gradual and systematic increase in training stress to maintain tissue overload and thus, stimulate continued training adaptation. | Progression rules should be clearly outlined in the exercise programme, that is, a detailed prescription of its realisation. If terminology such as 'as tolerated' is used, it should be clearly stated what is meant and how it is operationalised. | 'The duration of the exercise session is increased by 5 min per week'. 'The exercise intensity is increased by 0%–10% every 2 weeks based on the following progression rules…' |
| *Overload*: for training adaptations to occur, the exposure of tissues being trained must be exercise above the accustomed to training stress. | Rationale should be provided if exercise was prescribed adequately in relation to baseline testing or individual exercise capacity. For interventions longer than 6 weeks, reassessment should be performed during the intervention. | 'Exercise intensity was set at 80% of maximal capacity, which was retested every fourth week'. When overload (intensity) is solely prescribed on baseline testing in interventions longer than 6 weeks, this principle is not met. |
| *Specificity*: the closer the training stimulus resembled the characteristic of an activity (mode of exercise, energetic system, muscles trained), the better the training adaptations will be. | The purpose of the exercise training programme is specific and targeted to the patients' primary underlying problems known to establish the primary goal of interest. Moreover, outcome testing is performed in activity specific mode. | The intervention must be aligned with the aim. If the aim is to increase cardiovascular fitness, aerobic training should be used and not primarily strength training. This principle is not met if, for example, the efficacy of a walking intervention is evaluated with a cycling test. |
| *Individuality*: each individual will respond uniquely to the same training stimulus. Individuality refers to the modification of training to account for an individual's unique capacity and response to training. | A rationale should be provided on how the needs and abilities of any patient are translated in adaptations in individual training programmes. Preferably, the training is tailored based on individual training response. | Exercise is prescribed based on workloads corresponding to a per cent of peak workload (watt, speed and heart rate) obtained at baseline testing or intraintervention tests. Also, day-to-day variations in load are based on individual training response and not a generic rule. |
| *Initial values*: individuals with low initials values in the outcome of interest will show greater improvements compared with those with higher levels. | The programme is individually prescribed based on the results of each participant's baseline fitness assessment. Initial values should be obtained with an appropriate exercise test. | Participants with low levels of, for example, endurance, will be more likely to see a significant change than those with higher endurance levels. This principle is met if, for example, training load (frequency, duration or intensity) is based on initial values. |
| *Periodisation*: periodisation is characterised by a division of training into periods with specific goals and defined by the variation in training variables such as intensity, duration and frequency. | Rationale is provided for the systematic planning of training periods. Description of manipulation of different training components, for example, intensity, volume, frequency or recovery. | Example of linear periodisation description: 'For the first 4 weeks, the training target was 3 series of 6 min at 50% of pretraining maximum work rate. Intensity was increased with 5% when subjects completed all 3 series of cycling. When 3 series of 6 min cycling were again achieved, loads were increased. After 4 weeks of training, Wmax was reassessed and used for the remaining 6 weeks. Subjects performed five series of 3 min using 80% of the new Wmax'. |
| *Reversibility*: adaptations to exercise training is reversible, meaning that adaptations will return to baseline (detraining) if the training stimulus is removed. | A rationale is provided on how temporarily discontinuation is taken up in the training plan. | Interruptions in the intervention, due to illness, exacerbations or other reasons, must be considered in terms of return-to-training and how the training is recommenced. |

The principles of exercise training[26 27 38 39] are assessed in this systematic review.

We will provide a detailed account of the risk of bias for each study, contributing to the overall assessment of the validity and reliability of the evidence. This evaluation will aid in identifying potential sources of bias and in interpreting the findings of the included studies in the context of their methodological strengths and limitations.

### Synthesis and quality appraisal of training principles and quality of reporting

Quality appraisal related to our primary objective is performed independently by two reviewers, who are not involved in the included studies. A third reviewer will resolve disagreements and a majority rule will be used.

Given the nature of the data collected for our primary objective, no meta-analysis will be performed. Instead, we will synthesise the data as described below. The synthesis will be guided by relevant items from the PRISMA and synthesis without meta-analysis (SWiM) reporting guidelines. The SWiM[35] guideline is a nine-item checklist to promote transparent reporting of systematic reviews.

The variables listed under participant characteristics, aerobic training intervention characteristics, training principles and training variables will be rated as: '1' (reported), '0' (not reported), 'not clear', 'insufficient' or 'NA'.

The sum scores of training principles (up to eight points) and training variables (up to nine points) reported per study will be calculated. Training variables will be corrected for the number of NAs, since different numbers of variables might apply to different studies. The count of exercise training principles met within studies will be presented. The number and percentage of studies complying with all training principles and all relevant training variables will be calculated.

We will also critically appraise the included studies using the CERT checklist.[15] Similarly, each of the 19 items (16 items where items no. 7, 14 and 16 are divided into 'a' and 'b') in the CERT checklist will be reported as either '1' (reported), '0' (not reported), 'not clear' or 'NA'. As such, studies will be provided with 0–19 points on CERT-compliance. In addition, the items 'objective of aerobic training intervention', 'place of aerobic training intervention' and 'attendance' as listed under intervention characteristics will also be scored (0–3 points). Individual scores per paper will be presented in a supplementary material, while a summary score per variable is presented within the main text. The number and percentage of studies complying per CERT item will also be calculated. Tables will be used to present these data descriptively.

Publication details, reporting of participant and intervention characteristics will be narratively synthesised and presented using descriptive statistics to synthesise complete reporting. Summary scores will also be provided for participant characteristics.

## Meta-analysis

For our second objective, meta-analyses will be conducted to synthesise the efficacy of aerobic training interventions on selected outcomes. We are specifically interested in analysing the effects of application of training principles and complete reporting, as described under 'subgroup analysis and investigation of heterogeneity'.

Meta-analyses will be undertaken where meaningful; if the participants, type of intervention and study design are homogeneous enough to warrant pooling. We anticipate only continuous data to be reported for the outcomes of interest. Continuous data will be analysed as the mean difference (MD) where outcome data are reported on a uniform scale or standardised mean difference when different metrics are used but deemed clinically homogeneous. Meta-analyses will be based on change from baseline rather than postintervention values when available. Data extraction, and any needed imputations or will be done according to recommendations in Cochrane Handbook for Systematic Reviews of Interventions V.6.4. If needed, additional information from the study authors will be sought using e-mail. A random-effects model will be used to allow for random error and interstudy variability.

We will use per-protocol analyses as the effect of interest, where they are reported. If only intention-to-treat data is reported, this will be used for analysis. If both per-protocol and intention-to-treat data are reported, the intention-to-treat data will be used in a sensitivity analysis.

Where multiple trial arms are reported in a single study, we will include only relevant ones. If applicable, we will combine intervention arms or reported subgroups as per the recommendations in Cochrane Handbook for Systematic Reviews of Interventions V.6.4.

## Assessment of heterogeneity

Studies included in the review are likely to be methodologically diverse. As such, heterogeneity in the results is to be expected. We will use the $I^2$ and $\tau^2$ statistics to measure heterogeneity between the studies in each analysis. If we identify substantial heterogeneity ($I^2 \geq 50\%$), we will report it and explore the possible causes by prespecified subgroup analysis. The importance of the $I^2$ statistic will be interpreted together with the CI and p-value from the $\chi^2$ statistic, as well as the magnitude and direction of effect as described in the in Cochrane Handbook for Systematic Reviews of Interventions V.6.4.

## Assessment of reporting biases

Publication bias threatens the validity of a review's findings. To address this, if we can pool more than 10 studies, we will create and examine the asymmetry of a funnel plot and perform Egger's test to explore possible small study and publication biases. The funnel plot provides visual aid for detection of publication bias, indicating whether there is a lack of studies reporting negative or insignificant results.

To further investigate and adjust for publication bias, we will employ the trim-and-fill method.[36] This method is designed to identify and correct for asymmetries in the funnel plot that suggest the presence of publication bias. In the event of funnel plot asymmetry, we will use the trim-and-fill method to impute ostensibly missing effect sizes as a sensitivity analysis. Grading of Recommendations Assessment, Development and Evaluation (GRADE) is used for assessments of the quality of the evidence.

## Subgroup analysis and investigation of heterogeneity

We plan to conduct subgroup analyses to investigate the effect of application of training principles and complete reporting on intervention efficacy:

► Applying relevant principles versus not applying relevant principles: this subgroup analysis will be performed per principle.
► Complete or almost complete reporting of training variables (defined as reporting >85% of relevant variables, eg, eight out of nine variables) versus non-complete reporting of training variables (defined as reporting <85% relevant variables)
► High versus low compliance with CERT items. No accepted cut-offs exist for compliance to the CERT. Therefore, a post-hoc decision will be made on which cut-off to use to enable an adequate comparison with reasonable subgroups.
► Studies published after vs before publication of CERT.

Other planned subgroup analyses, not related to training principles or complete reporting but important

effect modifiers, which will be performed if an adequate number of studies per analysis is available:

► Sex.
► Exercise intensity.
► Duration of intervention (≥15 sessions compared with <15 sessions).
► Severity of COPD (GOLD A and B vs C and D or E).

Additional subgroup analyses may be added if an adequate number of studies per analysis is available. Any subanalysis that is not prespecified will be clearly presented as such and presented in a supplementary material.

To further investigate if methodological diversity moderates the effect on the selected outcomes, a meta-regression will be performed. A meta-regression is useful when there is a wide range of values in moderator variables but few studies with the exact same value for that moderator. We expect a wide range in moderator variables such as number of CERT items reported, or training principles applied.

We will employ a random-effects univariate meta-regression to examine the relationship between the extent of complete reporting, application of training principles and change in outcomes.[37]

The dependent variable in the meta-regression will be the MD in a continuous outcome (eg, change in 6MWD). We will examine how this effect estimate is influenced by explanatory variables, such as number of applied training principles or compliance with CERT (number of items). As such, the meta-regression model aims to predict the study's effect size, based on characteristics of the studies.

### Sensitivity analysis

We plan to carry out the following sensitivity analyses, removing the following from the primary outcome analyses to examine whether findings are robust to potentially influential decisions:

► Trials judged in the quality assessment to be of low quality.
► Trials with low or no compliance to training principles, defined as applying zero, one or two (out of eight) training principles.
► Trials reporting a low number of training variables (defined as reporting 33% or less of relevant variables).
► Trials with low compliance with the CERT checklist. No generally accepted cut-offs exist for compliance to the CERT. Therefore, a post-hoc decision will be made on which cut-off to use to enable an adequate sensitivity analysis.

We will compare results from a fixed-effect model versus a random-effect model. If we impute SD using correlation coefficients in the synthesis, sensitivity analyses will be performed with different values of the correlation coefficient to determine if the overall results are robust to the imputed correlation coefficients.[37] The sensitivity analyses will be presented in a summary table.

Other suitable sensitivity analyses, as identified during the review process, may also be performed, and reported.

### Assessment of the certainty of the evidence

We will employ GRADE approach to assess the certainty of the body of evidence related to the included primary and secondary outcomes. The GRADE approach includes assessing evidence across several key dimensions: risk of bias, imprecision, inconsistency, indirectness and publication bias. Thus, the quality of the evidence will be graded as 'high', 'moderate', 'low' or 'very low'. We will use the methods and recommendations described in the Cochrane Handbook for Systematic Reviews of Interventions V.6.4 using the GRADEpro software.

### Patient and public involvement

None.

### Ethics and dissemination

As this study is a systematic review, ethical approval is not required

The systematic review will be submitted for publication in a peer-reviewed journal. We also intend to present the findings at national and international conferences and inform patient organisations.

We will conduct the review according to this protocol and justify any deviations in a specific section of the systematic review results report.

### DISCUSSION

This systematic review is designed to critically assess the application of exercise training principles and the completeness of reporting in RCTs with an aerobic training intervention in the COPD literature. This is, to the best of our knowledge, the first systematic review addressing this topic in COPD trials.

The findings of this systematic review are expected to indicate shortcomings in aerobic training interventions and reporting of RCTs in COPD. Based on the findings, we will provide recommendations for future work to enhance the quality of exercise prescription as well as study quality and reporting practices. Identified limitations or strengths in the included studies could inform future guidelines revisions. This, in turn, might lead to more tailored and effective exercise training in PR.

Registration of the systematic review and publication of this protocol promotes transparency in the review process. It also minimises the risk of duplication of similar reviews. We strive for a high methodological standard by adhering to PRISMA-P and SWiM guidelines. This review is not without limitations. The exclusion of literature written in other languages than English might leave relevant literature out of the systematic review. For this reason, we will report the number of publications that are excluded due to this reason.

**Author affiliations**
[1]Community Medicine and Rehabilitation, Physiotherapy, Umea University, Umea, Sweden

[2]Department of Pulmonary Diseases, Radboud University Medical Center, Nijmegen, Gelderland, Netherlands
[3]Merem Pulmonary Rehabilitation Center, Hilversum, Netherlands
[4]Department of Pulmonary Medicine, Amsterdam University Medical Centres, Amsterdam, Noord-Holland, Netherlands

**Contributors** JJ contributed to the design of the study and the protocol, developed the data synthesis and meta-analysis, and wrote the initial version of the manuscript. AAFS contributed to the design of the study and the protocol, developed the search strategy, data extraction and data synthesis process. JJ and AS contributed equally as cofirst authors. HWHH contributed to conceptualising the study, the design of the protocol and reviewed the protocol critically for important intellectual content. JDB contributed to the design of the protocol, developed the data synthesis and meta-analysis and reviewed the protocol critically for important intellectual content. AN contributed to conceptualising the study, the design of the protocol, developed the data synthesis and meta-analysis and reviewed the protocol critically for important intellectual content as a cosenior author with PK. PK conceptualised the study, developed the data synthesis and meta-analysis, as the guarantor for the protocol and reviewed the protocol critically for important intellectual content. AN and PK contributed equally as cosenior authors. All authors read, approved and contributed to the final written manuscript.

**Funding** This work is undertaken as part of JJ's PhD work, funded by the Swedish Research Council (Vetenskapsrådet), grant number: 2020-01296. AS was financially supported by Lung Foundation (#5.1.18.232), Netherlands. Funders have had no role in the design of this protocol or in the systematic review; nor in the decision to report and publish the results. Non-financial support has been provided by the library at Radboud University, in developing the search strategy.

**Competing interests** None declared.

**Patient and public involvement** Patients and/or the public were not involved in the design, or conduct, or reporting or dissemination plans of this research.

**Patient consent for publication** Not applicable.

**Provenance and peer review** Not commissioned; externally peer reviewed.

**ORCID iD**
Johan Jakobsson http://orcid.org/0000-0002-9816-194X

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
