## [Reviewer comments · BMJ Open]

ARTICLE DETAILS

TITLE (PROVISIONAL)	Quality of aerobic training description and its relation to intervention efficacy in chronic obstructive pulmonary disease trials: study protocol for a systematic review, meta-analysis, and meta-regression
AUTHORS	Jakobsson, Johan; Stoffels, Anouk; van Hees, Hieronymus; De Brandt, Jana; Nyberg, André; Klijn, Peter

VERSION 1 – REVIEW

REVIEWER	Boyer, François Reims Champagne Ardenne University Hospital, Physical Medicine and Rehabilitation
REVIEW RETURNED	15-Feb-2024

GENERAL COMMENTS	Title too long ? meta-analysis on quality parameters ? Abstract ok Background clearly understood that the precise description of non-pharmacological intervention must be precisely detailed in the rational it would be useful to specify that the question of the effectiveness of aerobic training in COPD patients no longer needs to be demonstrated, but the modalities and principles for conducting efficient re-training could remain a central issue. The question is which clinical parameters can improve or reduce the effectiveness of an aerobic training program. Do we have a list of these parameters to propose? 124 “To which extent does RCTs with an aerobic training intervention in the COPD literature apply and report the principles of exercise training, and how well are the key training variables, intervention and participant characteristics reported in these trials? » ‘127 Another objective is to meta-analyse the results of aerobic training on exercise-related outcomes, and specifically to investigate if the completeness of reporting is related to efficacy of the interventions provided. As such, the following research question will be addressed:’ 130 « “Are the efficacy of aerobic training interventions related to the application of exercise training principles and/or a more complete description of key exercise training variables?” there are many questions that are best summarized for the reader
---

	if I've understood correctly, you're checking to see if the authors are indeed applying 'personalized' aerobic training (how?), by checking the parameters used and properly described? and you look at the effect of the principles of these interventions (personalized parameters) on the effectiveness of the aerobic training program? could you summarize the various stages of this work in a more synthetic phase or questions? What is the main question: the effectiveness of the aerobic program or the description of the parameters of the non-pharmacological program? I'm still stumped by the introduction because I don't quite understand whether we're interested in the efficacy parameters of an aerobic intervention or the efficacy of the programs, or the combination of the two, but it's difficult to do a meta-analysis? regarding the title, isn't it more about the efficiency of the description of aerobic training programs on its results than the observational description of the parameters of the care program? Methods precise and well described in its method insist more on metaregression, which is at the heart of the question raised by this protocol? Table, figure to synthesise metaregression Discussion OK limitations OK please improve abstract, introduction and explain better metaregression
--	--

REVIEWER	Burge, Angela T. Monash University, Respiratory Research@Alfred
REVIEW RETURNED	19-Feb-2024

GENERAL COMMENTS	Thank you for asking me to review this very comprehensive protocol which outlines a systematic review assessing the quality of aerobic training description in COPD trials and will provide an important contribution to this body of literature. BACKGROUND paragraph 1 I can see that the results of this review are intended to be applied in the context of pulmonary rehabilitation (after reading the discussion on page 17) but I am not sure that the naïve reader will immediately understand the role of exercise training within pulmonary rehabilitation (and references 2 and 3 refer to pulmonary rehabilitation [not exercise training per se]). The role of pulmonary rehabilitation in the non-pharmacological management of people with COPD, that exercise training is a critical component of pulmonary rehabilitation and that aerobic training is the most commonly investigated type of exercise training could be more clearly articulated to help the reader understand why this review is important. Additionally, references 3 and 6 were published in 2013 and do not represent the most up-to-date supporting citations for the relevant statements. METHODS With regard to the methods addressing the second objective, how were the secondary outcomes identified, and are the authors sure that this represents a comprehensive list of outcomes that may be
---

	used? Will studies be excluded if they report an outcome that is not listed e.g. from a step test? There are multiple inconsistencies with the most recent PROSPERO registration (20th Jan 2024) for e.g., please see study designs and interventions sections on page 6, risk of bias and study quality assessment on page 13.
--	---

VERSION 1 – AUTHOR RESPONSE

Reviewer: 1

Prof. François Boyer, Reims Champagne Ardenne University Hospital, Université de Reims Champagne-Ardenne UFR de Médecine

Comments to the Author:

Thank you for the thorough review and helpful comments to the manuscript.

Quality of aerobic training description in Chronic obstructive pulmonary disease trials: study protocol for a systematic review, meta-analysis, and meta-regression

- Title too long ? meta-analysis on quality parameters ?

While it might be on the longer side, we think the title is adequate including the overall study question and study design as per BMJ Open author guidelines

Abstract ok

Background clearly understood that the precise description of non-pharmacological intervention must be precisely detailed.

in the rationale it would be useful to specify that the question of the effectiveness of aerobic training in COPD patients no longer needs to be demonstrated, but the modalities and principles for conducting efficient re-training could remain a central issue.

We fully agree and have added this in the rationale. (p4, line 106-107).

The question is which clinical parameters can improve or reduce the effectiveness of an aerobic training program. Do we have a list of these parameters to propose?

124 “To which extent does RCTs with an aerobic training intervention in the COPD literature apply and report the principles of exercise training, and how well are the key training variables, intervention and participant characteristics reported in these trials?”

‘127 Another objective is to meta-analyse the results of aerobic training on exercise-related outcomes, and specifically to investigate if the completeness of reporting is related to efficacy of the interventions provided. As such, the following research question will be addressed:’

130 « “Are the efficacy of aerobic training interventions related to the application of exercise training principles and/or a more complete description of key exercise training variables?”

there are many questions that are best summarized for the reader

if I've understood correctly, you're checking to see if the authors are indeed applying 'personalized' aerobic training (how?), by checking the parameters used and properly described?

Indeed, there are two aims with two different research questions. We tried to simplify and clarify the Aims and objectives section (p5, line 153-162). The new text reads as follows:

We aim to investigate the application of exercise training principles and completeness of reporting in aerobic training interventions in the COPD literature. We also aim to investigate if efficacy of the interventions provided is related to the completeness of reporting and application of training principles. Our specific research questions are:

“To what extent does RCTs with an aerobic training intervention in the COPD literature apply and report the principles of exercise training, and how well are the key training variables, intervention and participant characteristics reported in these trials?”

“Is the efficacy of aerobic training interventions related to the application of exercise training principles and/or a more complete description of key exercise training variables?”

and you look at the effect of the principles of these interventions (personalized parameters) on the effectiveness of the aerobic training program?

This is indeed part of the second aim/research question. It is clarified in lines 153-162, page 5.

could you summarize the various stages of this work in a more synthetic phase or questions?

The section is revised

What is the main question: the effectiveness of the aerobic program or the description of the parameters of the non-pharmacological program?

Both are equally important, thus there is no main question. We have two aims/research questions. 1) We aim to investigate "the description of the parameters" as one of our aims and thereafter 2) we aim to investigate if "the efficacy of aerobic training interventions are related to the application of these parameters".

I'm still stumped by the introduction because I don't quite understand whether we're interested in the efficacy parameters of an aerobic intervention or the efficacy of the programs, or the combination of the two, but it's difficult to do a meta-analysis?

The introduction has been revised. The paragraphs on page 4, lines 100-113 tries to rationalize the first research question "To which extent does RCTs with an aerobic training intervention in the COPD literature apply and report the principles of exercise training, and how well are the key training variables, intervention and participant characteristics reported in these trials? For example "Inadequate reporting of aerobic training interventions has been reviewed previously in other chronic diseases¹⁹⁻²³, while trials in the COPD literature remains to be investigated."

The paragraphs on page 4-5, lines 115-150 thereafter aim to rationalize the second research question "Are the efficacy of aerobic training interventions related to the application of exercise training principles and/or a more complete description of key exercise training variables?" For example, "Evidence on the importance of incorporating the FITT-principles within aerobic training studies are starting to emerge and recent data suggests that FITT-principles such as frequency and time seem to be associated with the increase in peak oxygen uptake following aerobic training in individuals with COPD."

regarding the title, isn't it more about the efficiency of the description of aerobic training programs on its results than the observational description of the parameters of the care program?

It is about both, and we have slightly rephrased the title: "Quality of aerobic training description and its relation to intervention efficacy in Chronic obstructive pulmonary disease trials:..."

This makes it a few words longer but reflects the content more precisely.

Methods

precise and well described in its method

insist more on metaregression, which is at the heart of the question raised by this protocol? Table, figure to synthesise metaregression

While a meta-regression is an extension of a meta-analysis and sub-group analyses, we kept it in the title as we see it as a key analysis in the second objective. The section on the meta-regression have been updated (page 16, line 145-156).

Discussion OK

limitations OK

please improve abstract, introduction and explain better metaregression

We have made an attempt to improve these sections following your comments.

Reviewer: 2

Dr. Angela T. Burge, Monash University

Comments to the Author:

Thank you for asking me to review this very comprehensive protocol which outlines a systematic review assessing the quality of aerobic training description in COPD trials and will provide an important contribution to this body of literature.

We thank you for your valuable time and comments that have improved the manuscript.

BACKGROUND paragraph 1

I can see that the results of this review are intended to be applied in the context of pulmonary rehabilitation (after reading the discussion on page 17) but I am not sure that the naïve reader will immediately understand the role of exercise training within pulmonary rehabilitation (and references 2 and 3 refer to pulmonary rehabilitation [not exercise training per se]). The role of pulmonary rehabilitation in the non-pharmacological management of people with COPD, that exercise training is a critical component of pulmonary rehabilitation and that aerobic training is the most commonly investigated type of exercise training could be more clearly articulated to help the reader understand why this review is important. Additionally, references 3 and 6 were published in 2013 and do not represent the most up-to-date supporting citations for the relevant statements.

Thank you for this comment. We have tried to emphasize this a bit more in the first paragraph (p4, line 81-87) and the end of the introduction (p4, line 106-107). We have also added more up-to-date supporting citations, in addition to the 2013 statement.

METHODS

With regard to the methods addressing the second objective, how were the secondary outcomes identified, and are the authors sure that this represents a comprehensive list of outcomes that may be used? Will studies be excluded if they report an outcome that is not listed e.g. from a step test?

The outcomes were identified based on the preliminary searches made and review of the current literature on commonly used outcome measures for aerobic fitness within aerobic training studies and the final included outcomes were decided through discussion within our research constellation. Yes, if neither of the included outcomes is part of a specific study, it will not be included in the analysis for the secondary objective. We acknowledge that other outcomes could also be of relevance, but we are confident that the selected outcomes will provide a comprehensive overview.

There are multiple inconsistencies with the most recent PROSPERO registration (20th Jan 2024) for e.g., please see study designs and interventions sections on page 6, risk of bias and study quality assessment on page 13.

Indeed, the PROSPERO registration will be updated according to this recently updated design of the systematic review and the final comments by the reviewers.

VERSION 2 – REVIEW

REVIEWER	Boyer, François Reims Champagne Ardenne University Hospital, Physical Medicine and Rehabilitation
REVIEW RETURNED	12-Apr-2024
GENERAL COMMENTS	Thank you for the improvements to your research project, congratulations
REVIEWER	Burge, Angela T. Monash University, Respiratory Research@Alfred
REVIEW RETURNED	06-May-2024
GENERAL COMMENTS	I thank the author for their responses. My only remaining suggestion is that the search is updated prior to publication in order to reflect the most representative body of literature.

VERSION 2 – AUTHOR RESPONSE

We will indeed update the searches. The updated manuscript is adjusted in the abstract, line 190 "Timespan" and line 216 "search strategy".